# Properties of the Top Quark

Andreas Jung

Department of Physics & Astronomy, Purdue University, West Lafayette, IN 47906, USA;
andreas.werner.jung@cern.ch

**Abstract:** Recent measurements of the properties of the top quark at the CERN Large Hadron Collider are discussed. The results were measured for single and top quark pair production in their final states, including jets with either one or two leptons or only in hadronic final states. Top quark properties include angular correlations, top quark spin correlations, mass, and width. When looking towards the future, top quark properties open new and even interdisciplinary avenues for probing quantum information science.

**Keywords:** top quark properties; top precision frontier; top quark factory

## 1. Introduction

The top quark, denoted as $t$, holds a distinctive position in the Standard Model (SM) and its various extensions. As the heaviest elementary particle known, with a mass of around 173 GeV, it was first discovered in 1995 at the Tevatron $p\bar{p}$ collider through the efforts of the CDF and D0 collaborations [1,2].

The discovery of the Higgs boson in 2012 [3,4] made it even more important to measure the mass of the top quark accurately. The precise measurement of the top quark mass, when coupled with determinations of the masses of the W and Higgs bosons, now serves as a robust self-consistency check for the SM [5–9]. The top quark is particularly exceptional due to its large Yukawa coupling and its unique role in interactions with the Higgs boson. Studying these interactions is indispensable for understanding the extension of the SM and the model's behavior at higher energy scales or even the Planck scale. In particular, the relation between the masses of the top quark and the Higgs boson [10–13] is essential with regard to the stability of the electroweak vacuum.

The top quark has an exceptionally short lifetime, approximately $\tau \approx 5 \times 10^{-25}$ s [14], effectively precluding the formation of top quark hadrons under normal conditions. However, there exists a possibility for the creation of transient bound states, known as "toponium" [15]. The depolarization time frame for top quarks surpasses $\tau \approx 10^{-23}$ s [14], allowing for the direct observation of their intrinsic properties. This unique aspect facilitates the examination of various characteristics, such as the spin, spin correlations, and polarization of top quarks. By reconstructing their quantum spin state, researchers can delve into an intriguing aspect of quantum mechanics: entanglement.

This phenomenon is characterized by the quantum states of interacting particles remaining inseparable (entangled) irrespective of the distance or spacial separation between the particles. The measurement of entanglement in top quark events provides a new handle for exploring the fundamentals of quantum mechanics at higher energy scales and is connected to the realm of quantum information science in collider experiments [16,17].

The Large Hadron Collider (LHC) at CERN is a proton-proton (pp) collider operating at various energies, and it is renowned as a "top quark factory" due to the abundant production of top quarks. The predominant production mechanisms involve the creation of top quark–antiquark pairs ($t\bar{t}$) through the strong interaction ($gg$ / $q\bar{q} \rightarrow t\bar{t}$), serving as a fundamental test of quantum chromodynamics (QCD). Additionally, single top quarks are produced via the electroweak force, offering a means to test electroweak theory and

directly probe the Cabibbo–Kobayashi–Maskawa (CKM) matrix element $V_{tb}$. By utilizing all available production modes, measurements of top quark properties can be utilized to enhance our understanding of fundamental particles.

In the SM, the branching fraction for a top quark decaying into Wb is $\Gamma(Wb)/\Gamma(Wq) = 0.957 \pm 0.034$, with q = b, s, d [18]. The subsequent decay of the W boson into either $\ell\nu$ or $q\bar{q}'$, where $\ell$ represents an electron or a muon, is utilized to categorize $t\bar{t}$ events into decay channels. Namely, either both W bosons decay leptonically (dilepton final state), or only one W decays leptonically while the other W decays hadronically (lepton + jets or $\ell$ + jets final state), or none decay leptonically and both W decay hadronically (all-hadronic final state). Leptons in this categorization can also originate from semi-leptonic $\tau$ decays, while hadronic $\tau$ decays lead to additional jet activity in the event. The highest hadronic activity occurs in the all-hadronic final state consisting entirely of jets with jet multiplicities of up to 15 jets. The identification of jets originating from a b quarks can be achieved by utilizing the decay length of b quarks, which is in the order of $c\tau$ 10 s mm [18] and is commonly known as "b-tagging" [19,20]. It can be further improved through multivariate discriminant techniques employing a variety of variables [21]. These techniques involve a combination of variables describing the properties of secondary vertices and tracks with significant impact parameters relative to the primary vertex.

The specific final state particles of a $t\bar{t}$ pair can vary, and this is not only due to the decay channel and the number of quarks and gluons involved. For example, it can vary due to single or multiple leptons and one or more jets, some of which (or all) may be b-tagged, as well as combinations of the leptons and jets folded with missing transverse momentum due to the presence of neutrinos. Different decay channels, along with the number of top quarks produced, lead to a variety of final states, which provides valuable information for experimental analyses and allows for a comprehensive study of top quark properties.

This overview delves into a selection of measurements, providing an overview of the current landscape of experimental measurements of top quark properties and opens with a brief introduction in Section 2 on how to extract top quark properties. The measurements described in Section 3 encompass various aspects, such as angular correlations linked to asymmetries in top quark production and the correlation of the top quark's spin, which are detailed in Section 3.1. It is followed by a brief summary of Lorentz invariance violation in $t\bar{t}$ production. In Section 3.3, the overview delineates the present status of precision measurements of the top quark's mass, incorporating combined results from both ATLAS and CMS where applicable. Additionally, Section 3.4 discusses novel measurements to test the Yukawa coupling strength in top quark events, followed by results on the width of top quarks in Section 3.5. The review closes with Section 3.6, which explores the current state of measurements of top quark properties in associated production.

## 2. Accessing Top Quark Properties

In this context, what is commonly referred to as top quark properties encompasses measurements of various differential top quark cross-sections that allow (in a secondary step) for the extraction of the specific characteristics or properties (mass, spin, couplings, etc.) of the top quark. Figure 1 shows differential cross-section measurements at the LHC [22] in the $\ell$ + jets decay channels (see Figure 1), which provide excellent statistical power and are examples of distributions that will be used later on to extract top quark properties. A recent summary of measurement techniques and the results of differential cross-sections can be found in Ref. [23].

Examples include observations related to the polarization of top quarks and $t\bar{t}$ production level asymmetries at the LHC, but this is not the only way to measure top quark properties. Precise examination of these cross-sections is essential for thoroughly testing perturbative QCD (pQCD) predictions and identifying the potential signals of new physics within their theoretical frameworks. The meticulous measurement of top quark production cross-sections ensures a precise depiction of these processes, pinpointing areas for model refinement when necessary. This precision testing is vital for advancing our comprehen-

sion of top quark behavior, pushing the boundaries of the current theoretical framework, and facilitating more robust extraction of top quark properties. By scrutinizing the behavior of top quark pairs, these measurements contribute significantly to the validation and enhancement of our understanding of the fundamental physics involved, particularly in the domain of strong interactions.

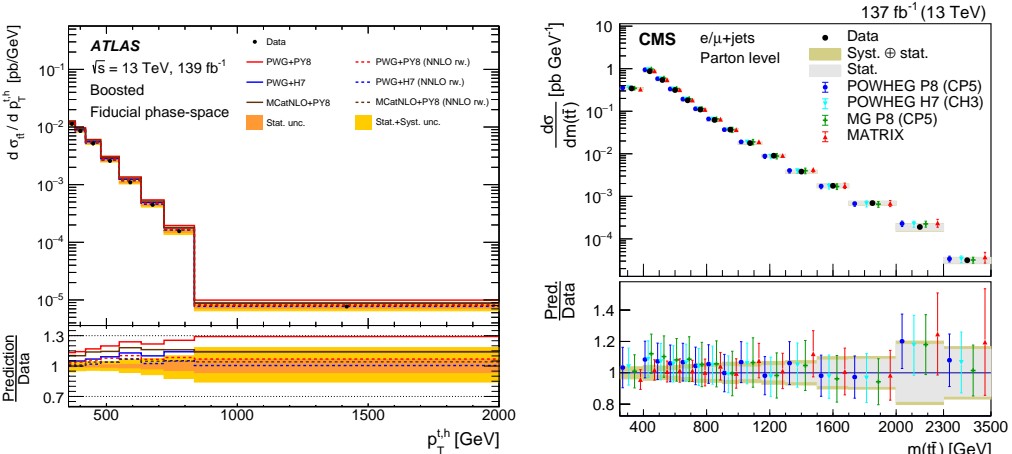

**Figure 1.** Differential cross-sections in the $\ell$ + jets channel as a function of the hadronic top quark $p_T$ (**left**) and of $m(t\bar{t})$ (**right**) [24,25].

The importance of single top quark production measurements cannot be overstated. At the LHC, the predicted cross-sections for single top quark production in the *t*- and *tW*-channels are not significantly smaller compared to the $t\bar{t}$ pair production cross-section. The number of single top quark events collected at the LHC is sufficient to make similarly precise and detailed studies of top quark properties. These measurements play a crucial role as essential tests for the electroweak theory, offering a unique avenue to probe the fundamental forces and interactions involved in these processes.

*Production of Top Quarks*

Accessing top quark properties in both single top and top quark pair production processes provides a comprehensive understanding of the fundamental characteristics of the heaviest elementary particle. In the context of single top quark production, the measured cross-sections in different channels, such as the t-(q'g → tqb̄) and tW-channels (gb → tW), offer unique opportunities for detailed studies. Despite the cross-sections not being significantly smaller than those for $t\bar{t}$ production at the LHC, single top events contain only one top quark, and the measurements serve as crucial tests for the electroweak theory. Measurements made with single top quark data have allowed for significant achievements, such as the analysis of the top quark-related components of the CKM matrix [26], the study of the polarization of individual top quarks, and leveraging single top quark events to constrain the structure of the proton, particularly by examining the ratio of the up-to-down quark content. These results are obtained via the measurements of the ratio between t and t̄ quarks.

Conversely, the production of pairs of top quarks, involving the creation of a top quark and its antiparticle, represents another crucial avenue for investigating top quark properties. The generation of $t\bar{t}$ pairs occurs through QCD processes involving strongly interacting colored gluons and quarks. Present-day theoretical predictions extend to the next-to-next-to leading order (NNLO), incorporating next-to-next-to leading log (NNLL) corrections in QCD, along with electroweak corrections at the next-to-leading order (NLO).

These predictions typically have relative uncertainties of 3.5% [27], including uncertainties related to renormalization and factorization scales, the proton parton density distribution functions (PDFs), and the determination of the strong coupling constant ($\alpha_S$).

Measurements in this context, often involving complex final states with multiple jets and leptons, contribute to a thorough examination of top quark characteristics, including their mass, spin, and decay properties. Measurements of single and top quark pair production cross-sections at the inclusive and multi-differential levels are described elsewhere in this issue.

### 3. Top Quark Properties in $t\bar{t}$ Production

The LHC produces an abundance of $t\bar{t}$ pairs, which allows for the scrutinization of their properties and behaviors with unprecedented precision. This unique capability transforms the LHC into an invaluable laboratory for unraveling the mysteries of the top quark and testing the predictions of the Standard Model. In particular, the vast amount of data allows for access to top quark properties, even at high masses for the $t\bar{t}$ pair ($m(t\bar{t})$) or large transverse momentum ($p_T$) scales, i.e., in the "boosted" phase space.

Measurements of $t\bar{t}$ production in the boosted regime offer a route to extracting top quark properties at high scales. This approach involves comparing measurements with predictions, leveraging the characteristics of boosted top quarks, which exhibit reduced contributions from bound state effects and Coulomb corrections, as well as lepton and trigger efficiency uncertainties that are much larger in the threshold region of $m(t\bar{t}) \approx 350$ GeV. However, studying boosted top quarks presents experimental challenges, requiring dedicated top quark taggers to maintain reconstruction efficiency.

In the analysis of boosted top quark events, the reconstruction of top quarks presents a particular challenge, especially when the lepton (either muon or electron) is non-isolated due to its close proximity to a b-jet. Hadronic decays of $\tau$ lepton decays can be exploited to study top quarks decaying into boosted $\tau$ leptons that are identified by using deep machine learning techniques [28].

Improved algorithms have been developed, resulting in better jet and lepton separation with higher efficiency for reconstructing the decay products of top quarks within the $\ell +$ jets channel, even when lepton isolation is not feasible. In such scenarios, the decay products of the hadronic W boson typically merge first, creating a "semi-resolved" topology. Advancing the measurements into the highly boosted phase space necessitates initial dedicated studies in this area to establish and refine the reconstruction methods [24,29]. Variables (see Figure 2) based on jet-substructure, i.e., n-subjettiness ($\tau_n$), allow for the SM to be challenged in a new phase space with reasonable avenues for extracting top quark properties.

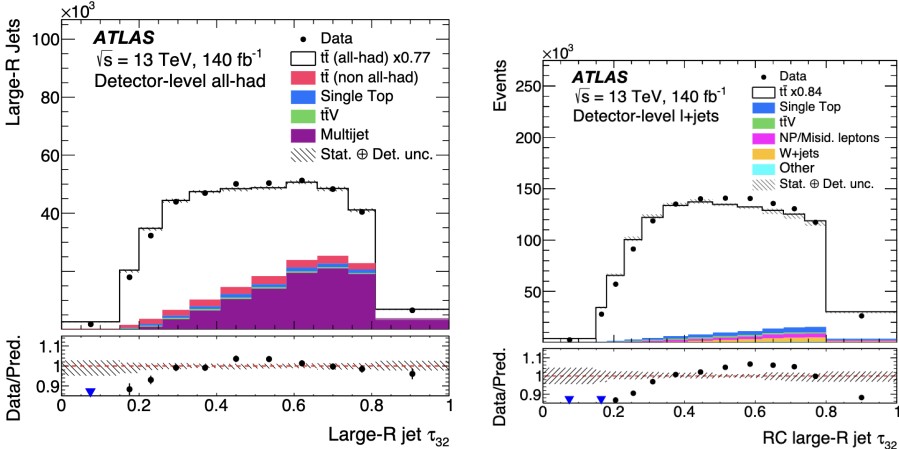

**Figure 2.** ATLAS measured detector level; $\tau_{32} = \tau_3/\tau_2$ distributions comparing in the (**left**) $\ell +$ jets and (**right**) all-hadronic channel for the data, the predicted $t\bar{t}$ signal, and the measured backgrounds [29].

### 3.1. Measurements of Angular Correlations

The measurement of correlations between the angles of various particles are sensitive probes with which to test the validity of the SM. Over the years, a variety of measurements have been carried out, such as production asymmetries, as well as measurements of top quark spin correlations and polarization. Typically, the angular correlations observed vary across the different center-of-mass energies during the LHC's operational phases. At the LHC, quarks tend to possess a higher average longitudinal momentum compared to anti-quarks. In the context of a pp collider, this distinction results in a broader rapidity distribution for $t$ production as opposed to the production of $\bar{t}$. Now, this allows for the definition of production level asymmetry employing the rapidity $y$ of a top and an anti-top quark. When using $\Delta|y| = |y_t| - |y_{\bar{t}}|$, the charge asymmetry $A_C$ is given by the following equation:

$$A_C = \frac{N(\Delta|y| > 0) - N(\Delta|y| < 0)}{N(\Delta|y| > 0) + N(\Delta|y| < 0)} \tag{1}$$

ATLAS and CMS have carried out measurements of charge asymmetries [30,31], including a joined analysis by ATLAS and CMS [32]. In the ensuing discussion, two of these findings are outlined. The current results from the LHC have not yet reached the level of precision of the charge asymmetry predicted by the SM at a value of $A_C = 0.0064^{+0.0005}_{-0.0006}$ [33]. Notably, ATLAS has reported evidence of a non-zero charge asymmetry [30], while not yet being able to verify the predicted SM value. CMS recently published a measurement of $A_C$ in the boosted phase space of $m(t\bar{t}) \geq 750$ GeV [31]. This measurement utilizes a binned maximum likelihood fit profiling systematic uncertainties that provide improved precision. In both scenarios, the measured top quark charge asymmetry is consistent with the SM prediction at NNLO in pQCD with NLO electroweak corrections. The charge asymmetry defined in Equation (1) can be modified to rely on leptons. This minimizes the uncertainties arising from reconstructing top quarks. Recently, ATLAS presented the first search for leptonic charge asymmetry in the $t\bar{t}$-associated production of W bosons [34] from initial state radiation. The ATLAS Collaboration reported an observed leptonic asymmetry of $-0.112 \pm 0.170(\text{stat}) \pm 0.054(\text{syst})$, which is in good agreement with theoretical predictions [33]. Although, this measurement presents a significant step forward in understanding the $t\bar{t}$ + W process, the measurement is still limited by statistical uncertainties. More data from LHC Run 3 (2022–ongoing) may substantially enhance this precision. This would also be useful in searches for physics beyond the SM. Looking even further ahead, future prospects involve extending the measurements of charge asymmetry by utilizing different variables, e.g., energy asymmetry [35]. Measuring top quark asymmetries with new variables and methods would refine our understanding of top quark production asymmetries and may help us identify subtle deviations.

The angular distributions of leptons resulting from top quark decays provide a precise measurement tool that is useful for investigating variables that are sensitive to the spin correlations and polarization of top quarks. Early measurements carried out at the Tevatron [36] revealed that the spin of top quarks is correlated, aligning with the predictions made by the SM. More refined measurements conducted at the all center-of-mass energies of the LHC [37–41] offer an unprecedented opportunity to scrutinize the SM at a detailed level. Recent measurements of the difference in azimuthal angle $\phi$ between the decay leptons, denoted as $\Delta\phi(\ell^+\ell^-)$, conducted by both ATLAS [41] and CMS [42], exhibit noteworthy agreement, as illustrated in Figure 3. However, when comparing these findings with SM predictions, a subtle tension becomes apparent. This tension is alleviated with higher-order corrections at NNLO [43] and by employing techniques that mitigate the impact of theoretical uncertainties on acceptance corrections [44].

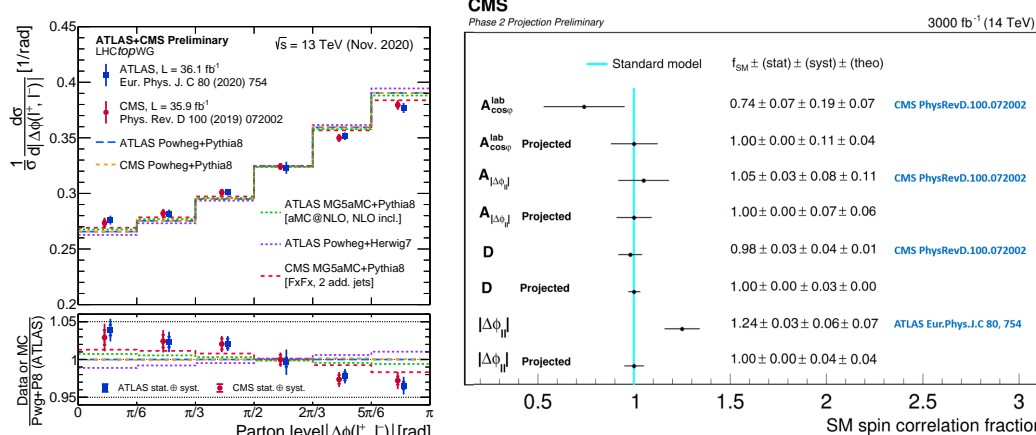

**Figure 3.** (**Left**): The measurements conducted by ATLAS [41] and CMS [42] on the angle of separation between the decay leptons from the tt̄ pair are set against the projections made by the SM. (**Right**): An estimated projection concerning the accuracy of future measurements of spin correlation and polarization of top quarks at the High-Luminosity LHC (HL-LHC) [45] and compared to existing measurements [41,42].

Current and future efforts at the LHC aim to enhance differential access to spin correlation and polarization information using full Run 2 data and the eagerly anticipated first Run 3 results. This approach seeks to better understand the evident mismodeling in the distribution of the opening angle of decay leptons, $\Delta\phi(\ell\bar{\ell})$, and of the top quarks, as highlighted earlier. A recent CMS projection evaluates the precision achievable in measuring various angular distributions with the data collected during the HL-LHC [45]. With an integrated luminosity of 3000 fb$^{-1}$ at $\sqrt{s} = 14$ TeV, providing a vast top quark data sample, statistical power becomes virtually limitless. Utilizing a generic future CMS detector simulation through DELPHES [46] offers an initial assessment of the expected precision in measuring the strength of spin correlations within the SM. The parameter $D$, derived from the differential helicity angle (cos $\varphi$) distribution, is expected to have a precision of better than 3% [45]. Figure 3 (right) displays the outcome of this study, including other variables, and this is compared to existing measurements by ATLAS and CMS using partial Run 2 data. The precision of top quark spin correlation variables is leveraged to project the sensitivity to supersymmetric top quark partners (stops) in the degenerate phase space [41]. The latter is defined as the phase space where the stop mass equals the mass of its decay particles (top quark and neutralino).

*3.2. Search for Lorentz Invariance Violation*

A test of Lorentz invariance violation (LIV) can be achieved by a detailed examination of tt̄ pair production with the $pp$ collision data collected by the CMS detector at the LHC, operating at a center-of-mass energy of $\sqrt{s} = 13$ TeV. The study, motivated by the possibility of deviations from Lorentz invariance, builds upon the first of such investigations at the Tevatron [47], but it utilizes vastly more data. Data in the dilepton channel ($e\mu$ only) were selected for this study and correspond to an integrated luminosity of 77.4 fb$^{-1}$ [48]. Specifically, the analysis explores the differential normalized cross-section for tt̄ production as a function of sidereal time, introducing LIV as an extension of the SM—see Figure 4. In the context of an effective field theory, the predictions include the modulation of the tt̄ cross-section with sidereal time when LIV is present. The investigation extracts bounds on LIV couplings, revealing compatibility with Lorentz invariance, with an impressive absolute precision of 0.1–0.8%.

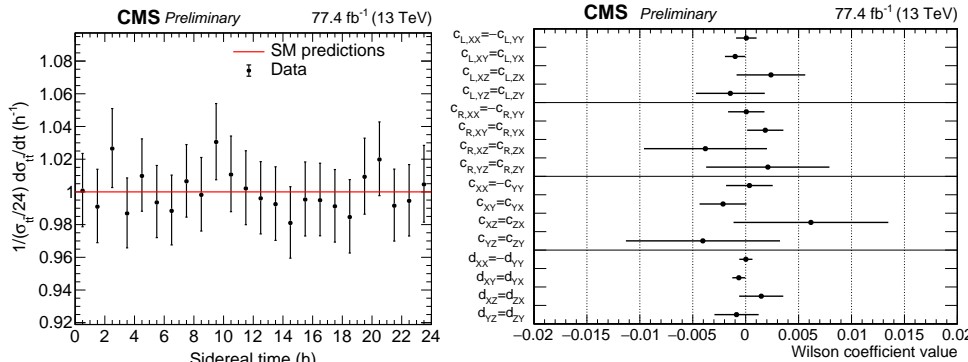

**Figure 4.** (**Left**): CMS data in t$\bar{\text{t}}$ production is shown as a function of siderial time and is compared to the SM expectation of no dependency at all [48]. (**Right**): Limits on EFT parameters, implementing LIV in t$\bar{\text{t}}$ production [48].

This search not only delves into potential LIV in top quark production but also serves as a precision test of special relativity with top quarks. Notably, the precision achieved in this study surpasses previous measurements [47] in this domain by two orders of magnitude, underlining the significance of this comprehensive exploration.

### 3.3. Top Quark Mass

Various measurements of the top quark mass, $m_t$, have been conducted at the LHC, and while it's impractical to cover all the details here, we focus on the latest and most relevant findings. Figure 5(left) presents a summary (as of June 2023) of top quark mass measurements utilizing kinematic or so-called "direct" methods. Both ATLAS [49] and CMS [50] experiments have measured the top quark mass with typical uncertainties well below 0.5 GeV in absolute values, which is below 0.3% in relative uncertainty. These measurements span dilepton, $\ell$ + jets, and all-hadronic decay channels. The Tevatron has also contributed, with the latest combination [51] and an initial world combination [52].

The world average, as depicted by the vertical grey band in Figure 5 (left), typically has a relative uncertainty of around 0.5%. This precision is somewhat diminished compared to the more recent results from the LHC, a discrepancy that is mainly due to the lack of updated LHC data in the world combination [52]. Historically, combinations have employed the BLUE method [53–55], but recent developments in measurement techniques, specifically those using profile likelihood approaches, have led to the introduction of a more precise likelihood method [56], which is now commonly adopted for LHC top quark combinations.

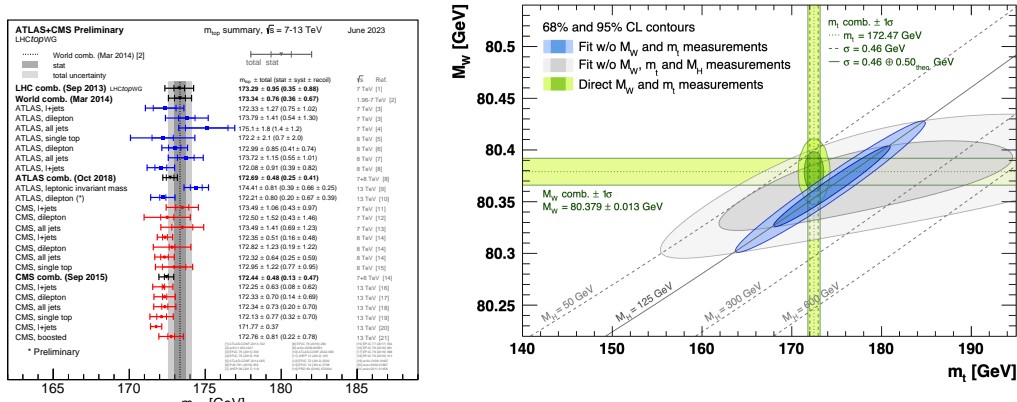

**Figure 5.** On the (**left**): Determinations of the top quark mass based on direct measurement techniques [57], with the following Refs. [50,52,58–73]. On the (**right**): The $m_t$ versus $m_W$ plane, displaying the GFitter global fit outcome derived from top quark mass results [9].

Top quark mass measurements at LHC Run 2 confirm the LHC as a top quark factory, providing ample data for precise determinations. Most recently, such a measurement was provided by CMS using profile likelihood methods with five observables [72]. Here, top quark mass determination is based on calculating the invariant mass of three jets originating from the same top quark, which includes one b jet and two light jets from the W boson. In order to reconstruct the complete t$\bar{\text{t}}$ system, a minimization of a $\chi^2$ is carried out, taking into account the anticipated resolutions and combinatorial factors. The analysis incorporates observables, offering optimal constraints on the top quark mass and dominant uncertainties, including those impacting the jet energy scale. Figure 6 (left) illustrates the step-wise improvement in total precision by progressively adding these observables. Figure 6 (right) presents additional observables as inputs to the ML fit, along with their post-fit probability density functions. This analysis notably addresses the treatment of statistical fluctuations affecting systematic uncertainty modeling directly in relation to the likelihood, a crucial consideration for future analyses with increased data and smaller uncertainties.

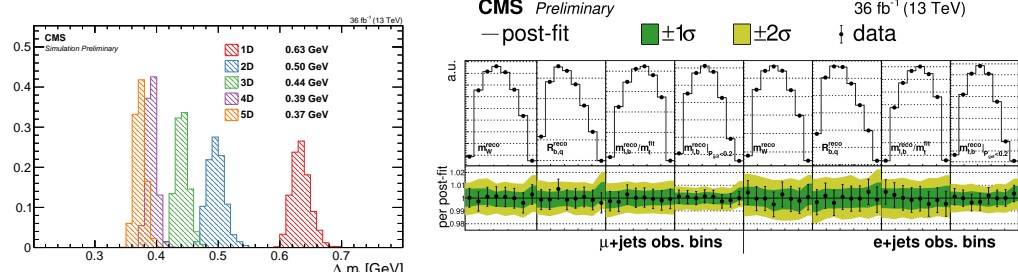

**Figure 6.** On the (**left**): The capability of the ML techniques to constrain uncertainties for extracting $m_t$ is achieved by incorporating additional observables [72]. On the (**right**): The input to the ML fit comprises additional observables, along with their post-fit probability density functions [72]. The 1 sd and 2 sd uncertainty bands are depicted by the green and yellow bands, respectively.

The stability of the electroweak vacuum serves as a powerful test of the SM up to energy scales close to the Planck scale, which is also relevant for cosmology since many processes in the early universe could have triggered a decay of the electroweak vacuum [74]. In the context of particle physics, the electroweak vacuum state is closely tied to the Higgs field and its potential. A crucial question revolves around whether the universe resides in the absolute lowest energy state or if the current vacuum state is metastable, potentially capable of transitioning to a more stable state. Figure 5 (right) displays the $m_t - m_W$ plane, where $m_t$ is determined via a combination of ATLAS and CMS measurements by the GFitter collaboration [9], and this is updated in Ref. [75]. This combination incorporates the latest measurements and is subject to an additional theoretical uncertainty of 500 MeV, addressing the "controversy" on whether the MC mass (e.g., in PYTHIA) equals the pole mass (used in fixed order calculations) in the field of top quark physics [76]. This controversy, gaining renewed attention with the precision of LHC measurements beyond 0.3%, necessitates a precise evaluation of the variances and uncertainties associated with this conversion, as outlined in a recent comprehensive review article on the subject referenced in Ref. [76].

The interplay between the masses of the top quark, the W boson, and the Higgs boson [3,4] serves as a robust self-consistency test for the SM, contributing insights into the stability or potential meta-stability of the SM vacuum [13]. Ongoing discussions, particularly regarding the W boson mass measurement, underscore the dynamic nature of our understanding in this fundamental realm. A recent revision of the W boson mass measurement by CDF [77] diverges notably from the global average. The community is actively deliberating the implications of this result [75]. Current measurements, coupled with theoretical extrapolations at NNLO in the SM, suggest a scenario where the vacuum is either meta-stable or, intriguingly, positioned precisely at the threshold between stability and meta-stability.

A substantial push towards a better understanding of how top quark mass measurements and their uncertainty relate between ATLAS and CMS is presented in Ref. [78]. A comprehensive combination of 15 top quark mass measurements from the ATLAS and CMS experiments at the LHC is presented—see Figure 7 (left). The datasets utilized cover integrated luminosities of up to 5 and 20 fb$^{-1}$ for pp collisions at center-of-mass energies of 7 and 8 TeV, respectively. This combination encompasses measurements in top quark pair events involving both $\ell$ + jets and hadronic top quark decays, as well as a measurement from events enriched in single top quark production via the electroweak t-channel. The resulting combined top quark mass is $m_{\rm t} = 172.52 \pm 0.14$ (stat) $\pm 0.30$ (syst) GeV, with a total uncertainty of 0.33 GeV or better than 0.2%.

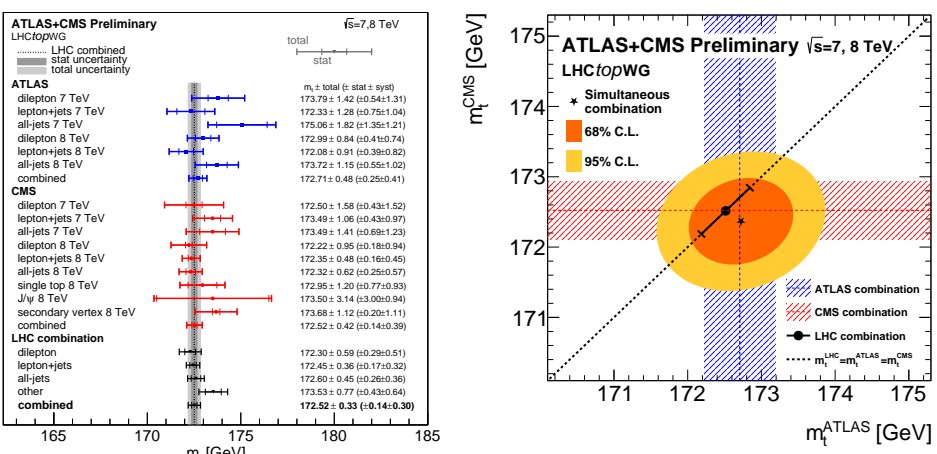

**Figure 7.** On the (**left**): The most recent LHC combination integrates 15 top quark mass measurements from various decay channels conducted by both ATLAS and CMS [57]. On the (**right**): The correlation between the measured $m_{\rm t}$ values in ATLAS and CMS is examined and compared to the combined value from the LHC [78].

ATLAS and CMS simultaneously extract the top quark mass based on ATLAS measurements, $m_{\rm t}^{\rm ATLAS}$, and based on CMS measurements, $m_{\rm t}^{\rm CMS}$, using a BLUE combination of 15 input measurements—see Figure 7 (right). The ellipses show the 68% and 95% confidence intervals, indicating good agreement between $m_{\rm t}^{\rm ATLAS}$ and $m_{\rm t}^{\rm CMS}$. The observed correlation between them is 0.15. The central values and uncertainties for individual ATLAS and CMS combinations are represented, and the full marker denotes the central value of the LHC combination ($m_{\rm t}^{\rm LHC}$), assuming equality with $m_{\rm t}^{\rm ATLAS} = m_{\rm t}^{\rm CMS}$. This latest combination of top quark mass measurements has a high precision of 0.2% thanks to a substantially improved understanding of the correlations between systematic uncertainties in either set of mass measurements by ATLAS or CMS. Achieving advancements in reducing uncertainties necessitates collaborative work from the entire scientific community. Advancements are being actively pursued on both experimental and theoretical sides through the collaborative efforts of the LHC Top Working Group (LHCtopWG) [57], exemplifying a concerted drive towards scientific progress in this arena. This group serves as a hub for researchers to collectively contribute to the ongoing developments in understanding and refining our knowledge of top quark physics.

Alongside direct measurements, alternative methods for determining the top quark mass aim to enhance precision by complementing systematic uncertainties or improving the understanding of well-defined renormalization schemes [79]. A prevalent alternative technique involves extracting the top quark mass from the $t\bar{t}$ production cross-section, which measures the top quark pole mass in a direct approach [22,80]. While Tevatron had limitations, the higher LHC data statistics enable not only inclusive but also multi-differential techniques to extract the top quark mass precisely. The abundance of data

allows for the simultaneous fit of $\alpha_S$ and the top quark mass, achieving uncertainties below 0.9 GeV, e.g., even in the boosted phase space using a "jet mass" proxy [81].

Alternative methodologies for estimating $m_t$ focus on different features of event data. An example involves the use of the invariant mass of a muon from the W boson's leptonic decay in conjunction with a soft muon found within the b jet. This technique presents a measurable variable sensitive to the top quark mass [82]. The reliance on muons in this method reduces the vulnerability to uncertainties typically associated with the jet energy scale and the modeling of the top quark production process. Nonetheless, this approach introduces its own set of uncertainties, primarily stemming from the nuances of B hadron decay processes, including the details of B fragmentation and decay branching fractions. The effectiveness of this approach in constructing a top quark mass-sensitive observable is demonstrated in Figure 8, which showcases the post-fit distribution based on the selection of muon pairs with the same sign.

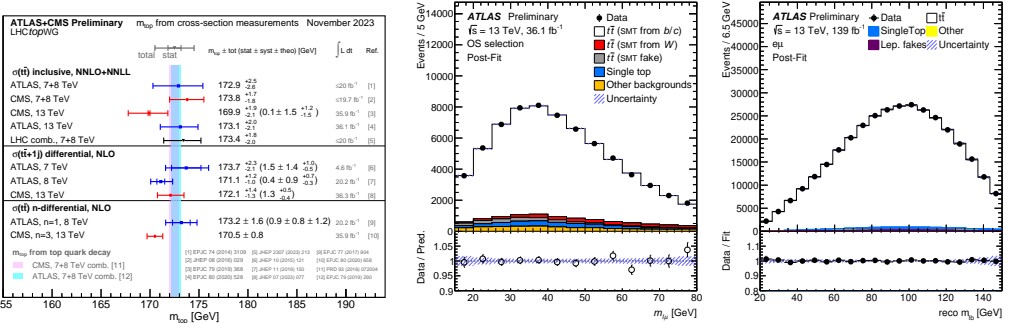

**Figure 8.** (**Left**): An overview of the measurements of the top quark mass by ATLAS and CMS, derived from observables in $t\bar{t}$ production [57] (**Middle**); details provided in Refs. [22,44,50,80,83–88]: The distribution after fitting, using pairs of muons with the same sign, is utilized to establish an observable sensitive to $m_t$ [82]. (**Right**): A description of the observables employed in estimating the width of the top quark. The uncertainties at the post-fit level are determined using the correlation matrix derived from the fitting process [89].

### 3.4. Top Quark Yukawa Coupling

The interactions between the top quark and the Higgs boson field are governed by the Yukawa coupling strength ($y_t$). The top quark is the one fermion that has the highest Yukawa coupling strength. The measurements of its value can shed light on the role of the top quark in the mechanism of electroweak symmetry breaking. CMS verified the strength of the top quark Yukawa coupling by analyzing the kinematic distribution of top quark pairs [90], using dilepton events from $pp$ collision data and the full CMS Run 2 dataset of 137 fb$^{-1}$. The mass of the $t\bar{t}$ system and the rapidity difference of the top quark and antiquark are particularly sensitive to $y_t$. The measurement yields a value of $y_t = 1.16^{+0.24}_{-0.35}$, constraining $y_t < 1.54$ at the 95% confidence level.

### 3.5. Top Quark Width

The precise calculation of the top quark width is achievable in the SM [91,92], and deviations could suggest new physics. Indirect measurements relying on SM couplings lack model independence since they assume SM couplings. Pioneered at the Tevatron [93,94] and refined at the LHC Run I [95], direct top quark width measurements, such as the recent one by ATLAS [89], employ width-sensitive final state distributions, utilizing the invariant masses of the lepton and b-tagged jet, $m(lb)$, and the $b\bar{b}$ pair, $m(b\bar{b})$—these allow for the constrainment of the uncertainties arising from JES uncertainties. The measured top quark width, $\Gamma_t = 1.9 \pm 0.5$ GeV, aligns with the SM prediction.

### 3.6. Top Quark Properties in Associated Production

In the initial stages of Run 2, the precise measurements of inclusive cross-sections in the associated production of vector bosons with $t\bar{t}$ pairs have provided important results to better understand the SM. These processes include $t\bar{t}$ + W, $t\bar{t}$ + Z [96], and $t\bar{t}$ + photon, offering access to enhanced charge asymmetry, Z boson coupling to $t\bar{t}$ pairs, and the determination of the top quark's electric charge. These channels also serve as crucial backgrounds for beyond the SM (BSM) searches, especially in cases involving the associated production of additional $q\bar{q}$ pairs, such as $b\bar{b}$, which contribute significantly to the $t\bar{t}$ pairs associated with the Higgs boson. Additionally, rare channels like tZq events, where a single top quark is produced with a vector boson (Z boson and a quark or jet), have been observed by ATLAS [97] and differentially measured by CMS [98]. CMS's measurements, including the Zq -to-Z$\bar{q}$ rate ratio, offer a unique avenue to explore another property of the top quark: spin asymmetry $A_\ell$. It is defined as $A_\ell = 1/2Pa_\ell$ with $P$ being the polarization of the top quark and $a_\ell \approx 1$ [99] being the spin analyzing power (see Figure 9(left)). The ratio of the Zq-to-Z$\bar{q}$ rates gives access to the polarization of the top quark [98] and Figure 9 (right) provides an overview of the associated cross-section measurement that can be utilized to extract properties once sufficient data is available. An overview of ATLAS and CMS measurements of t$X$ with ($X$ =Z or $\gamma$) cross-sections at 13 TeV and comparisons to NLO QCD theoretical calculations is displayed in Figure 10(left). The latest results include the observation of a single top quark with a photon by ATLAS [100] and evidence for the production of the tWZ process by CMS [101]. The different phase space regions used for the measurements are denoted as "Vis 1", "Vis 2" and "Vis 3", and they are highlighted for the ATLAS and CMS tq$\gamma$ measurements. Mild tensions between the experimental results and the theoretical predictions can be seen for some of these rare processes.

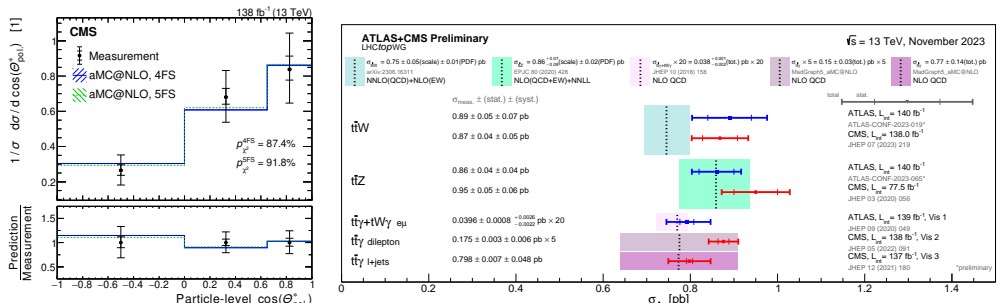

**Figure 9.** (**Left**): The spin asymmetry, determined through the cross-sections of top quark production in association with a Z boson, is contrasted with SM predictions [98]. (**Right**): A comprehensive overview of the associated production of vector bosons at the LHC [57] - details can be found in Refs. [41,102–107].

The top quarks produced in association with other bosons, such as photons, Higgs, or Z bosons, can be utilized to determine flavor-changing neutral currents (FCNCs). Figure 10 (right) displays the summary of LHC results for FCNCs, including comparisons with various new physics models, assuming all other FCNC processes are negligible. The limits are presented as top quark decay branching ratios, considering both FCNC top quark decay and production vertices in some cases. Both ATLAS and CMS have a variety of FCNC searches in this sector, with the most recent ones using the full Run 2 data briefly discussed here. ATLAS uses the full Run 2 data of 139 fb$^{-1}$ to explore FCNCs involving a photon and a top quark [108]. No significant event excess is observed over the background prediction, and the 95% confidence level upper limits are placed on the strength of left- and right-handed FCNC interactions. FCNC interactions can also manifest themselves in events involving the top quark, the Higgs boson, and an up-type quark (q=c,u), where no significant excess is observed by ATLAS in the t→ qH (H→ $\gamma\gamma$) process [109]. Most recently, CMS submitted a search for FCNCs in events with a photon and additional jet activity [110] and new Higgs bosons [111], while ATLAS set stringent limits for FCNCs in

events with heavy Higgs bosons [112]. In addition to the existence of an FCNC in the top quark sector, CMS has looked into charged lepton flavor violation involving trilepton final states. The observed data align with SM expectations, and the results have been employed to extract 95% confidence level upper limits for Wilson coefficients—more details are in Ref. [113].

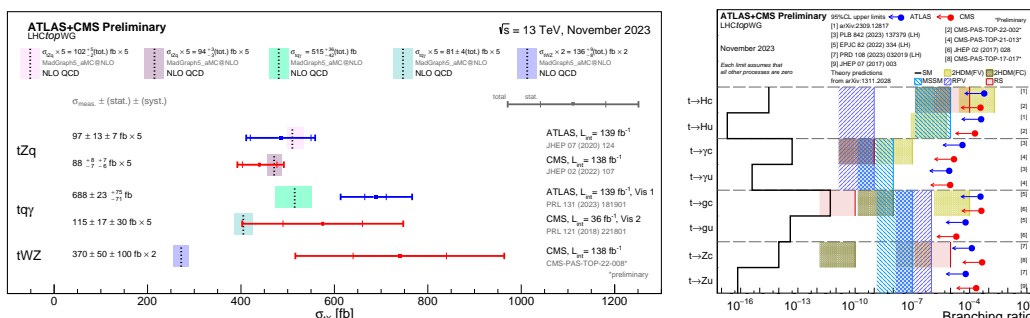

**Figure 10.** (**Left**): Summary of t*X* cross-section results by ATLAS and CMS [57] with additional information being provided in Refs. [97,98,100,101,114]. To facilitate visualization on a consistent scale, the results for tZq and the CMS tqγ measurements are multiplied by a factor of 5, while those of the CMS tWZ measurement are multiplied by a factor of 2. The theory bands encompass uncertainties arising from renormalization and factorization scales, as well as from PDFs. (**Right**): Summary of 95% confidence level observed limits on top quark decay branching ratios via FCNCs to a quark and a neutral boson (t→ *X*q, where *X*=g, Z, γ or H; q=u or c) by the ATLAS and CMS collaborations [57]; additional information is provided in Refs. [108,109,115–122].

The exploration of four top quark production [123–125] represents one of the most intriguing and, until recently, uncharted territories at the LHC. Current limits, as illustrated in Figure 11(left), present intriguing indications of a potential cross-section enhancement in comparison to the SM prediction of $\sigma(t\bar{t} + t\bar{t}) = 12$ fb [126]. This process, involving the simultaneous production of four top quarks, serves as a valuable avenue for probing the top quark Yukawa coupling [90], as depicted in Figure 11 (right). This underscores the pivotal role of four top quark productions in advancing our understanding of top quark interactions and potentially uncovering new physics phenomena [127].

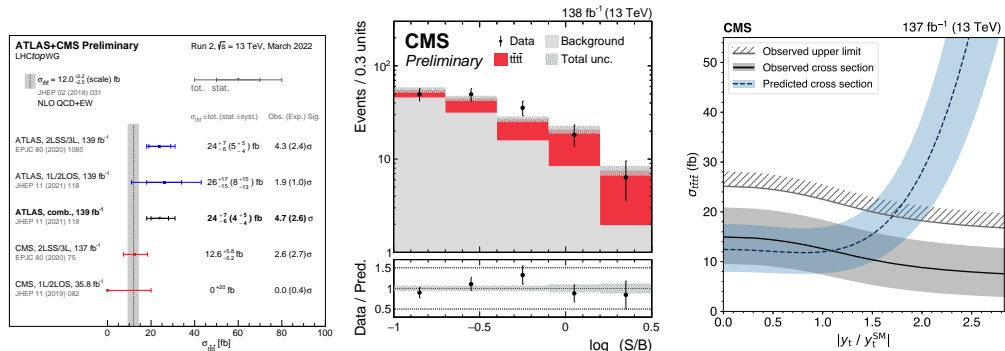

**Figure 11.** (**Left**): The production cross-section of $t\bar{t} + t\bar{t}$ as measured by ATLAS and CMS is compared with the theoretical prediction from the SM - further information is provided in Refs. [123,124,128,129]. (**Middle**): CMS's observation of the $t\bar{t} + t\bar{t}$ production cross-section [127]. (**Right**): The intervals of a 95% confidence level concerning the top quark Yukawa coupling, deduced from the observed $t\bar{t} + t\bar{t}$ cross-section [123–125].

The observation of the production of four top quarks in *pp* collisions marks a significant achievement, drawing from a dataset collected by the CMS experiment [127]. This milestone is based on an integrated luminosity of 138 fb$^{-1}$. The analysis focuses on events featuring two same-sign, three, and four charged leptons (electrons and muons), along

with additional jets. By employing sophisticated multivariate discriminants to discern the signal process from predominant backgrounds, the measured signal cross-section is $17.9^{+3.7}_{-3.5}$ (stat) $^{+2.4}_{-2.1}$ (syst) fb, aligning closely with the best available theoretical predictions. The observed (expected) significance of the signal stands at 5.5 (4.9) standard deviations above the background-only hypothesis.

## 4. Summary

In the past decade, LHC experiments have revolutionized our understanding of top quark properties, challenging state-of-the-art theoretical predictions. Recent measurements, such as top quark spin correlations, reveal the mild tension with the Standard Model (SM), mitigated by higher-order corrections. Angular correlations, such as charge asymmetries, hint at deviations from the SM. Notably, the latest ATLAS + CMS combination of top quark mass measurements reaches a precision of 0.2%. Overcoming systematic uncertainties demands community-wide efforts, emphasizing unified signal modeling in community efforts such as the LHCtopWG. Ultimately, the HL-LHC promises a dataset exceeding 1 billion top quarks, which, in the context of top quark properties, will be a significant challenge while promising exciting new results, and this will surely also see the use of cutting-edge techniques not currently widely utilized in top quark physics.

Excitingly, top quark properties open new avenues for probing quantum information science, forming a new field of quantum observables in top quark physics and unlocking exciting opportunities for new physics discoveries.

**Funding:** This research was funded by Department of Energy award number DE-SC0007884.

**Data Availability Statement:** Data are contained within the article.

**Conflicts of Interest:** The author declares no conflicts of interest.

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
