# Peer review of "Properties of the Top Quark"

_universe, doi:10.3390/universe10030106_

Round 1
Reviewer 1 Report
Comments and Suggestions for Authors
In the overview paper entitled “Properties of the Top Quark” by Andreas Jung, the properties of the top quark measured in pp collisions at LHC are discussed. This overview is useful for the community of high energy physics and can be published after some revisions.
Some comments
The rapidity and transverse momentum spectra of both single tops and top quark pairs produced in pp collisions at LHC are also needed, if available.
You may add the selected conditions such as pp, energy, rapidity range, transverse momentum range in all plots to show a clearer picture.
Many abbreviations are not given their full names when they first appear, eg. L31, LHC; L36, CKM; L104, NNLO, NNLL; L105, NLO; L107, PDF.
L39, L181, L326, L344, “the Standard Model (SM)” is repeatedly explained after L9.
L168, if NNLO is explained in L304, you may change “next-to-next-to-leading order (NNLO)”to “NNLO”.
Author Response
Dear Reviewer 1
thanks a lot for the careful read of the draft - please find below answers line by line for your comments. These should address your comments with the updated manuscript.
The rapidity and transverse momentum spectra of both single tops and top quark pairs produced in pp collisions at LHC are also needed, if available.
--> The differential cross sections in single and pair top production are discussed elsewhere in the larger review on top quarks.
You may add the selected conditions such as pp, energy, rapidity range, transverse momentum range in all plots to show a clearer picture.
--> Unfortunately these plots are official CMS and ATLAS plots and can not be changed by individual people, even if I am an author of CMS. The only way to access that information is by looking into the provided references.
Many abbreviations are not given their full names when they first appear, eg. L31, LHC; L36, CKM; L104, NNLO, NNLL; L105, NLO; L107, PDF.
--> A clear oversight, thanks for paying attention to these and abbreviations have been properly introduced now.
L39, L181, L326, L344, “the Standard Model (SM)” is repeatedly explained after L9.
--> Good catch, these have been removed now.
L168, if NNLO is explained in L304, you may change “next-to-next-to-leading order (NNLO)”to “NNLO”.
--> fixed now.
Reviewer 2 Report
Comments and Suggestions for Authors
Dear author,
It was a pleasure to read this nice review on the current status of the measurements of the top-quark properties. I only have a few minor suggestions for your consideration, which can be found below.
Kind regards
--------
general on content: maybe you could briefly mention the results on top quark flavour changing neutral currents? I understand you might not want to expand on this, but a few sentences and the corresponding references could be of interest to the reader.
general on style: I suppose that units (e.g. fb^-1) should be in roman, not italic
ll.19 and 22: it would be helpful to the reader to give references for the livetime and depolarization time of the top quark.
ll.40-42: I find the sentence "The subsequent decay (...)" a bit confusing. Maybe you could rephrase it?
l.44: I suggest to add a reference to b-tagging
l.70: "Rigorous examination" -> maybe something like "precise determination"?
l.81: I suggest to change "significance" to "importance" or similar, since significance has usually a precise statistical meaning
l.120: it might not be obvious to the reader what "high scales" means in this context.
l.175: the phi here is the azimuthal angle?
ll.218: "more accurate approximate" reads a bit oddly, in my opinion.
ll.345 and forward: the vacuum stability results are, above all, a consistency test of the SM rather than a profound statement on the nature of the universe. In l.249 you also mention this. I suggest you introduce the topic with this focus.
ll.243-244: MC mass is not only a pythia parameter, but rather one in any MC
Figure 7, right. I believe there is a more recent version of this plot: https://twiki.cern.ch/twiki/bin/view/LHCPhysics/LHCTopWGSummaryPlots#Associated_tt_X_Production
l.330: it could be helpful to the reader to add relevant references to "new physics phenomena" in this context.
Ref 16: null, null should be fixed.
Ref 33: has a strange formatting for the equations
Ref 42: Maybe something is missing, since "; .;" appears?
In some references the author is missing (e.g. ref 48, 63 and 66)
Author Response
Dear Reviewer 2
thanks a lot for the careful read of the draft - please find below answers line by line for your comments. These should address your comments with the updated manuscript.
general on content: maybe you could briefly mention the results on top quark flavour changing neutral currents? I understand you might not want to expand on this, but a few sentences and the corresponding references could be of interest to the reader.
--> added now
general on style: I suppose that units (e.g. fb^-1) should be in roman, not italic
--> fixed now.
ll.19 and 22: it would be helpful to the reader to give references for the livetime and depolarization time of the top quark.
--> thanks for the catch, fixed now!
ll.40-42: I find the sentence "The subsequent decay (...)" a bit confusing. Maybe you could rephrase it?
--> indeed, that’s a very long one. Broken up and made more clear now.
l.44: I suggest to add a reference to b-tagging
--> indeed, added standard ones and some ML enhanced ones from ATLAS and CMS.
l.70: "Rigorous examination" -> maybe something like "precise determination"?
--> indeed, not ideal word choice – used precise.
l.81: I suggest to change "significance" to "importance" or similar, since significance has usually a precise statistical meaning
--> fixed.
l.120: it might not be obvious to the reader what "high scales" means in this context.
--> explained in more detail and fixed.
l.175: the phi here is the azimuthal angle?
--> yes, added now to the text for clarity.
ll.218: "more accurate approximate" reads a bit oddly, in my opinion.
--> turned around and now reads “introducing a more accurate (less approximate) likelihood method”
ll.345 and forward: the vacuum stability results are, above all, a consistency test of the SM rather than a profound statement on the nature of the universe. In l.249 you also mention this. I suggest you introduce the topic with this focus.
--> I agree - in order to be consistent with the earlier introduction that part is now altered in the same focus.
ll.243-244: MC mass is not only a pythia parameter, but rather one in any MC
--> absolutely, an oversight…fixed now and added a new reference by Hoang et al. as well.
Figure 7, right. I believe there is a more recent version of this plot: https://twiki.cern.ch/twiki/bin/view/LHCPhysics/LHCTopWGSummaryPlots#Associated_tt_X_Production
--> an oversight, thanks for the catch. Added the new plot.
l.330: it could be helpful to the reader to add relevant references to "new physics phenomena" in this context.
--> added now too.
Ref 16: null, null should be fixed.
--> thanks – fixed!
Ref 33: has a strange formatting for the equations
--> thanks, fixed!
Ref 42: Maybe something is missing, since "; .;" appears?
--> did not see an issue in the bib entry – I will check with the journal and its style file, some issues I could not fix despite best intentions.
In some references the author is missing (e.g. ref 48, 63 and 66)
--> "null, null" and others have been fixed where possible. Results not yet published but submitted and electronically available are appearing with no authors in the style of this journal. I have inquired with the journal already in the past how to fix this and they agreed to add the usual CMS or ATLAS authors and collaboration info.
Reviewer 3 Report
Comments and Suggestions for Authors
Author Response
Dear Reviewer 3
thanks a lot for the careful read of the draft - please find below answers line by line for your comments. These should address your comments with the updated manuscript.
I had some trouble in converting and answering line by line to the comments in the attached pdf since my mac did not want to turn this into plain text format. I hope the attached pdf works and my answers are coherently matched to the relevant comment.
thanks for the careful read and the clear improvements to the draft!
best

Reviewer 4 Report
Comments and Suggestions for Authors
Author Response
Dear Reviewer 4
thanks a lot for the careful read of the draft - please find below answers line by line for your comments. These should address your comments with the updated manuscript.
Comments and suggestions:
1- General: As the aim of the article is to provide a summary of the most recent
measurements of the important properties of the top quark, there are several interesting
and important properties from ATLAS and CMS which have not been discussed.
Specially, the following topics related to the top quark properties make sense to be
added: (1) CPT test (mt-mt_); (2) top quark Yukawa coupling from the top pair
kinematic distribution; (3) top quark pole mass measurement extracted from the ttbar
cross section; (4) top quark decays through Flavor Changing Neutral Currents
(FCNCs): tqZ, tqH, tqg, and tqGamma. A short paragraph about the SMEFT probes
within CMS and ATLAS would be also informative for the readers to be included.
--> Thanks for the suggestions, indeed a review is never fully comprehensive. To broaden the review a short section on the yukawa coupling from cross sections (latest ones) has been added now. Extractions of the pole mass have been mentioned in the text but no citations existed, except for the one on jet mass by CMS. The updated version has now references for ATLAS and CMS pole mass extractions from cross sections. In addition, also FCNCs have been mentioned in the associated production section while aspects of new physics in FCNC or SMEFT are covered in other parts of this larger review and hence are not added in detail to the properties review (the usual summary figure is in now though)
2- Line 19: to be more precise, it would be good to write the lifetime as 5 × 10!"#.
--> done.
3- Line 40: as we are moving to high precision, it is better to mention the accurate number
of branching fraction of the tàWb (for instance the PDG average).
--> indeed, this is fixed now!
4- Line 106: a reference for the indicated uncertainty of 3.5% for the cross section needs
to be added.
--> added the top++ where this is taken from.
5- Line 141: a short definition of the charge asymmetry at the LHC is useful for readers
here.
--> added the relevant equation and more text to introduce it better.
6- Line 147: instead of 0, the SM prediction with its theoretical uncertainty is more useful.
--> done now and added the reference too.
7- Line 148: Reference [18] superseded by JHEP 2308 (2023) 077.
--> fixed now.
8- Line 209: Figure 3(a) à Figure 3 (left).
--> thanks – fixed.
9- Lines 317-318: Both ATLAS and CMS have measurements of SM production of
tqGamma. As the production of tqZ has been mentioned, it would be good for
completeness to add SM tqGamma results.
--> Indeed, an oversight – thanks! Added a tX summary figure, citations and a brief capture of current state of affairs.
10- There are problems in References such as refs. [9], [16], [42], [63].
--> "null, null" and others have been fixed where possible. Results not yet published but submitted and electronically available are appearing with no authors in the style of this journal. I have inquired with the journal already in the past how to fix this and they agreed to add the usual CMS or ATLAS authors and collaboration info.